# The Therapeutic Potential of Low-Intensity Pulsed Ultrasound in Enhancing Gallbladder Function and Reducing Inflammation in Cholesterol Gallstone Disease

**DOI:** 10.3390/bioengineering12010034

**Published:** 2025-01-04

**Authors:** Fang Chen, Run Guo, Tian Chen, Liping Liu, Fan Ding, Gang Zhao, Bo Zhang

**Affiliations:** 1Department of Ultrasound, Shanghai East Hospital, School of Medicine, Tongji University, Shanghai 200120, China; 2133320@tongji.edu.cn (F.C.); grun@tongji.edu.cn (R.G.); chentian109@163.com (T.C.); liuliping10202021@163.com (L.L.); 2Center of Gallbladder Disease, Shanghai East Hospital, School of Medicine, Tongji University, Shanghai 200120, China; dingf1991@163.com; 3Institute of Gallstone Disease, School of Medicine, Tongji University, Shanghai 200331, China; 4State Key Laboratory of Cardiology and Medical Innovation Center, Shanghai East Hospital, School of Medicine, Tongji University, Shanghai 200120, China

**Keywords:** cholesterol gallstone disease, gallbladder emptying, low-intensity pulsed ultrasound, CCKAR, smooth muscle

## Abstract

Background: Cholesterol gallstone disease (CGS) is often accompanied by gallbladder contraction dysfunction and chronic inflammation, but effective therapeutic options remain limited. This study investigates whether a low-intensity pulsed ultrasound (LIPUS) treatment can improve gallbladder motility and alleviate chronic inflammation while exploring the underlying mechanisms. Methods: Gallbladder motility was assessed through in vitro and in vivo contraction tests, while bile condition was evaluated by observing bile crystal clearance. Tissue analysis and Western blotting were performed to examine the expression of the cholecystokinin A receptor (CCKAR) and α-smooth muscle actin (α-SMA) as markers of gallbladder smooth muscle health and the inflammatory microenvironment. Blood cholesterol levels were measured via biochemical assays. Results: LIPUS treatment obviously enhanced gallbladder contractility in response to CCK-8 stimulation and accelerated bile crystal clearance. It also reduced inflammatory cell infiltration and tissue edema, and promoted new capillary formation in the gallbladder, mitigating the progression of CGS. Furthermore, LIPUS restored CCKAR expression and improved the thickness of the gallbladder smooth muscle layer, providing a structural basis for increased smooth muscle contractility. Conclusion: LIPUS improves gallbladder motility and reduces chronic inflammation in CGS by enhancing CCKAR expression and smooth muscle integrity. These findings highlight the potential of LIPUS as a non-invasive therapeutic approach for managing CGS.

## 1. Introduction

As a non-invasive therapeutic ultrasound, the acoustic intensity of low-intensity pulsed ultrasound (LIPUS) is less than 3.0 W/cm^2^, typically non-thermal (including cavitation and biological effects), and capable of exerting therapeutic effects without apparently increasing biological temperature [1]. Studies have demonstrated its effectiveness in enhancing muscle function and reducing inflammation [2,3,4].

Cholesterol gallstone disease (CGS) is a common biliary condition characterized by the presence of stones in the gallbladder or bile ducts, with gallbladder stones being the most prevalent, affecting approximately 10% of the population in China [5]. Gallbladder stones can be classified into three main types based on their composition: cholesterol stones, bile pigment stones, and mixed stones. Cholesterol stones are the most common, accounting for approximately 80% of all gallbladder stones [6].

Bile primarily consists of cholesterol, phospholipids, and bile acids, which collectively maintain a balanced state of bile metabolism. A long-term high-cholesterol and high-fat diet causes an imbalance in cholesterol secretion and metabolism in bile, leading to excessive cholesterol precipitation [7]. Some free cholesterol and phospholipids form unilamellar vesicles, which merge to form multilamellar vesicles. The accumulation of crystals into nuclei serves as a precursor to stones [8]. Furthermore, excessive cholesterol secretion deposits on the plasma membrane surface of the gallbladder wall can result in decreased membrane fluidity, damage to the smooth muscle, and impaired gallbladder contraction [9]. This mechanism is crucial for the development of cholesterol gallstone disease.

Following gallbladder stone formation, the gallbladder wall becomes rough due to prolonged friction from the stones, which can lead to chronic inflammation or even gallbladder cancer. Consequently, decreased gallbladder contractile function and gallbladder wall chronic inflammation are important pathological factors in the development of cholesterol gallstone disease.

This study established a guinea pig model of cholesterol gallstone disease as a basis for LIPUS irradiation and measurement of gallbladder contractility to assess contractile function. Histological techniques, including hematoxylin-eosin staining (HE), Masson trichrome staining, and transmission electron microscopy (TEM), were employed to evaluate pathological status and cell structures, while microscopy assessed changes in bile crystal density. Additionally, immunohistochemistry (IHC) and Western blotting (WB) were utilized to assess the expression levels of α-SMA and CCKAR.

## 2. Materials and Methods

### 2.1. Animal Models and Grouping

Four-week-old male guinea pigs (weighing 250–300 g) were purchased from Shanghai Jiesijie Laboratory Animal Co. Ltd. (Shanghai, China). Male animals were selected with the intention of minimizing the effect of sex-specific estrogen on experimental cholesterol gallstone formation, as we were unaware of a previous study that demonstrated no influence of sex-hormones on gallbladder motility [10]. All guinea pigs were housed in the Laboratory Animal Center of East Hospital, Tongji University (Shanghai, China) with a 12-h light–dark cycle (ambient temperature of 24 °C and humidity of 55%) and had an adaptive feeding schedule for one week. The Animal Ethics Board of the University of Tongji Medical School approved all experimental protocols (approval number: TJBB05924Z01). The animals were anesthetized with isoflurane gas (induction anesthesia: concentration of 5%; maintenance anesthesia: concentration of 1.5–2.5%; RWD Life Science, Shenzhen, China), and all animals were euthanized by cervical dislocation at the end of the experiment. A cholesterol cholelithiasis guinea pig model was constructed using a high-cholesterol, high-fat diet for 8 weeks containing 2% cholesterol, 0.5% bile salts and 15% butter [11,12]. The guinea pigs were randomly divided into four groups, including a normal control group (NC), gallstone group (GS), LIPUS sham treatment group (LIPUS(−)), and LIPUS treatment group (LIPUS(+)), with nine animals in each group, as follows: NC: given normal feed for 8 weeks; GS: given lithogenic feed for 8 weeks; LIPUS(−): guinea pigs with CGD that received 4 weeks of the LIPUS sham treatment; LIPUS(+): guinea pigs with CGD that received 4 weeks of the LIPUS treatment. The LIPUS therapy instrument was purchased from Shenzhen Dongdixin Technology Co., LTD. (Shenzhen, China), and the parameters were adjusted to an intensity of 0.8 W/cm^2^, a frequency of 3 MHz, and a duty cycle of 30%.

### 2.2. Tissue Preparation

All the guinea pigs were required to fast for 12 h prior to the experiment. Then each guinea pig in the NC, GS, LIPUS(−), and LIPUS(+) group was briefly induced with anesthesia using isoflurane, the guinea pigs were fixed supine on an experimental plate, and the abdominal hair was shaved. The abdominal organs were then exposed by sterile minitype surgical instruments, and the gallbladder was precisely removed for three sections: one that was immediately placed in Krebs–Hensleit’s solution (KHS) for muscle strip contractility capacity, another saved in 4% polyfluoroalkoxy (Wuhan Servicebio Technology, Wuhan, China) for HE/Masson staining and the IHC experiment, and a third that was stored directly in a freezer at −80 °C for protein quantitative analysis. Fresh gallbladder bile was extracted using a syringe. Additionally, we collected 2 mL of blood from the heart.

### 2.3. Gallbladder Muscle Contractility Study In Vivo

A PHILIPS ultrasonic instrument (Washington, DC, USA) equipped with an 18-MHz 3D array probe (V6-5) was used to determine the 3D gallbladder volume. After the guinea pigs were fasted for 12 h, anesthetized with isoflurane, and their skin was prepared, 3D volume images of the gallbladder with an empty stomach were collected. Then the octapeptide cholecystokinin (CCK-8; dose: 0.1 mg/mL; 0.06 μg/mg) was injected intramuscularly into the right thigh gastrocnemius muscle [13], and 8 min after the injection, 3D volume images of the gallbladder were again acquired to calculate the gallbladder emptying fraction (GBEF) with EPIQ7 GI 3DQ software (QLAB 12.0). GBEF was calculated based on the gallbladder volume (mL) change after the injection to assess the gallbladder contraction capacity. Statistical analyses were based on the GBEF induced by CCK-8, where GBEF = [|(response volume − control volume)|/control volume].

### 2.4. Gallbladder Muscle Contractility Study In Vitro

We collected gallbladder muscle strips (10 mm × 3 mm) from the NC, GS, LIPUS(−), and LIPUS(+) groups and suspended the strips in organ baths filled with 20 mL of KHS with the following composition (mg): 280 of anhydrous CaCl_2_, 350 of KCl, 290 of MgSO_4_-7H_2_O, 2100 of NaHCO_3_, 160 of KH_2_PO_4_, 6920 of NaCl, and 2000 of D-glucose, all of which was fully dissolved in ddH_2_O and fixed to 1 L. The temperature of the bath was maintained at 37 °C and consistently aired with 95% O_2_ and 5% CO_2_. The gallbladder muscle strip was attached to the force transducer (AD Instruments, Sydney, NSW, Australia) in the organ bath. Each muscle strip was allowed 40 min of balance time for a preload of 1.5 g. The immediate effects of cholecystokinin octapeptide (CCK-8; 5 μmol/L; Aladdin, Shanghai, China) on gallbladder tone were detected. The mean preload level was recorded as the control value, while the effect level of CCK-8 was the response value. Statistical analyses were based on the CCK-8-induced change rate of muscle strip tension (R), where R = [|(response value − control value)|/control value].

### 2.5. Observation of the Density of Bile Crystals

We collected fresh bile from the gallbladder in sample tubes. We added 1 μL of bile dropwise directly to the slide, and bile crystals were observed with a polarized light microscope (Olympus U-POT, Shinjuku, Japan).

### 2.6. Determination of Serum T-CHO, LDL-C, and HDL-C

Blood samples of each guinea pig were collected in a centrifuge tube, then stewed for 2 h at 37 °C to allow the serum to layer before transferring the samples to the centrifuge tube to spin at 4000 rpm for 15 min at 4 °C. The supernatant was then extracted to the corresponding labeled sample tubes for storage. Serum T-CHO, LDL-C, and HDL-C values were measured using an automated biochemical analyzer and corresponding kits.

### 2.7. Detection of α-SMA and CCKAR Expression by Immunohistochemistry

The gallbladder specimens were paraffin embedded. Then, we proceeded to IHC stain the samples using the following antibodies: anti-α-SMA (1:200, MyBioSource, San Diego, CA, USA) and anti-CCKAR (1:200, ABclonal Technology, Wuhan, China). Before adding the primary antibodies, non-specific antibodies were blocked with 3% bovine serum albumin (MyBioSource, San Diego, CA, USA). Then, corresponding horseradish peroxidase-conjugated secondary antibodies (Wuhan Servicebio Technology, Wuhan, China) were used with the corresponding samples. Finally, H_2_O_2_ and 3,3-diaminobenzidine tetrahydro chloride (Wuhan Servicebio Technology, Wuhan, China) as the chromogen were used for antibody-positive localization. Three fields at ×20 magnification per slice were obtained randomly for CCKAR density detection, and another three were obtained randomly for the thickness of smooth muscle detection.

### 2.8. Protein Extraction and Western Blot Analysis

Radio-immunoprecipitation assay lysis buffer (Beyotime Biotech, Shanghai, China) was applied to extract the gallbladder total protein. Protein concentrations were calculated using a bicinchoninic acid protein concentration measurement kit (Beyotime Biotech, Shanghai, China). Additionally, 10% sodium dodecyl sulfate polyacrylamide electrophoresis gels (Millipore, Burlington, MA, USA) were used for protein band separation, then the sample band was transferred to a polyvinylidene fluoride membrane (Millipore, Burlington, MA, USA). Then the membranes were cropped into strips according to the molecular weight of the target proteins, and to block non-specific binding sites, the target protein bands were blocked released with a quick blocking buffer (Beyotime Biotech, Shanghai, China) at room temperature (RT) for 20 min. The anti-α-SMA (1:1000, MyBioSource, San Diego, CA, USA), anti-CCKAR (1:1000, ABclonal Technology, Wuhan, China), and anti-α-Tubulin (1:500, Invitrogen, Carlsbad, CA, USA) primary antibodies were incubated overnight at 4 °C. The next day, after the overnight incubated protein strips were washed 3 times by Tris-buffered saline with Tween-20, the strips were bathed with the horseradish peroxidase-conjugated secondary antibodies for 1 h at RT. An Enhanced Chemiluminescence Plus chemiluminescence reagent kit (Vazyme Biotech, Nanjing, China) was used for visualization. Image J software (Image J 1.53, NIH, Bethesda, MD, USA) was applied for quantification. α-Tubulin stayed as an internal control for the results normalization.

### 2.9. TEM

Selected fresh gallbladder tissue pieces (3 mm × 3 mm) were put into Eppendorf tubes with fresh 2.5% glutaraldehyde (Wuhan Servicebio Technology, Wuhan, China) at 4 °C for fixation and preservation. Before examination, the tissues were washed with phosphate-buffered saline 3 times for 15 min each. Then, the samples were fixed with 1% OsO_4_ under dark conditions (pH 7.4) for 2 h at RT. After removing the OsO_4_ and dehydrating the gradient, resin was added to embed the samples. The resin blocks were cut 60–80 nm thin with an ultra-microtome and fished out onto 150-mesh cuprum grids with formvar film. The gallbladder tissues were stained by 2% uranyl acetate and 2.6% lead citrate, and then the tissue ultrastructure was observed and photographed under TEM. (The technology was supported by Wuhan Servicebio Technology, Wuhan, China).

### 2.10. Statistical Analysis

GraphPad Prism 9.0 (GraphPad, San Diego, CA, USA) was used for date analysis, and each experiment was repeated three times. Results are presented as a mean value ± standard deviation (SD). Statistical differences between groups were either analyzed with a two-tailed Student’s *t*-test or one-way analysis of variance, if the data were normally distributed. The Shapiro–Wilk test and the Brown–Forsythe test were used to check whether the samples were normally distributed and to determine the homogeneity of variance in the samples, respectively. The normality test of the residual was realized by the Shapiro–Wilk test. If not, either the Mann–Whitney or Kruskal–Wallis test was used. A *p* < 0.05 was considered statistically significant. “Appendix A” for one example of Normality test and analysis of variance.

## 3. Results

### 3.1. A High-Fat Diet Causes Cholesterol Gallstones

Guinea pigs were fed a high-fat diet or a normal diet for 8 weeks. The polarized light microscopy analysis of the bile revealed a clear increase in stone crystal density in the high-fat diet group, indicating a higher abundance of stone components, whereas the normal diet group exhibited almost no stone crystals in their bile (Figure 1A). Additionally, serum biochemical tests indicated that the levels of total cholesterol, low-density lipoprotein cholesterol, and high-density lipoprotein cholesterol were apparently higher in the high-fat diet group compared to the normal diet group (respectively, *p* < 0.0001, *p* = 0.0010, *p* = 0.0011, Figure 1B–D); serum T-CHO, HDL-C, and LDL-C values: NC vs. GS:T-CHO: 1.09 ± 0.24 mM vs. 6.58 ± 1.54 mM, HDL-C: 0.09 ± 0.01 mM vs. 1.50 ± 0.83 mM, and LDL-C: 0.90 ± 0.18 mM vs. 2.81 ± 1.11 mM. These findings suggest that a high-fat diet leads to a sharp increase in serum cholesterol levels, which in turn raises the density of stone crystals in the bile, ultimately promoting the formation of gallstones.

### 3.2. LIPUS Effectively Inhibited the Formation of Cholesterol Gallstones

We subjected guinea pigs on a high-fat diet to regular LIPUS treatment for four weeks. After treatment, we analyzed bile samples using polarized light microscopy. The results indicated that the density of bile salt crystals in the treatment group was decreased compared with that in the sham treatment group (Figure 2A), suggesting that the LIPUS treatment effectively promoted the dissolution of bile salt crystals. However, the serum cholesterol levels revealed no differences in the treatment and sham groups: T-CHO: 2.41 ± 0.94 mM vs. 3.04 ± 0.87 mM, HDL-C: 0.15 ± 0.05 mM vs. 0.13 ± 0.05 mM, LDL-C: 2.293 ± 0.50 mM vs. 1.86 ± 0.37 mM with *p* = 0.1580, *p* = 0.0542 and *p* = 0.5024, respectively (Figure 2B–D).

### 3.3. LIPUS Alleviates Damage to the Gallbladder Caused by High Cholesterol-Saturated Bile

To investigate the potential mechanism by which LIPUS inhibits bile crystal formation, HE and Masson staining analyses were conducted on gallbladder samples from both the treatment and sham groups. The Masson staining results show that there is no difference in gallbladder wall fibrosis between the treatment group and the sham group (Figure 3C), indicating that supersaturated bile has caused chronic damage to the gallbladder wall. The results of the HE staining indicated that inflammatory infiltration in the gallbladder wall of the treatment group was obviously reduced compared to the sham group, with notable improvements in inflammatory cell infiltration and tissue edema, and a decrease in newly formed capillaries (Figure 3A,B right). This suggests that LIPUS treatment effectively reduces the inflammatory response in the gallbladder, indicating that LIPUS may protect the structure and function of gallbladder tissues by alleviating damage to the gallbladder wall.

TEM was employed to examine gallbladder samples and assess damage to smooth muscle cells and Cajal cells. The results indicated that, in the sham treatment group, mitochondria within Cajal cells of the gallbladder wall exhibited irregular swelling, the endoplasmic reticulum was markedly swollen, and the nuclear membrane displayed signs of contraction (Figure 3D left). Meanwhile, the morphology of smooth muscle cells exhibited apparent deformation, characterized by sparse synaptic structures and the nearly complete disappearance of intracellular reticular structures (Figure 3E left). This suggests that, under conditions of supersaturated bile acids, gallbladder wall cells experienced damage, indicating structural injury. In contrast, damage to smooth muscle cells and Cajal cells in the LIPUS irradiation group was apparently less severe than in the sham treatment group. This finding suggests that the LIPUS irradiation treatment reduced the extent of damage caused by supersaturated bile acids to gallbladder wall cells, preserving their structural integrity and functional state, thus promoting the recovery of the gallbladder’s physiological function. These results provide crucial experimental evidence for understanding the mechanism of LIPUS in relieving cholesterol gallstone disease and suggest that LIPUS may reduce the impact of supersaturated bile on gallbladder contraction by modulating the gallbladder wall microenvironment and alleviating tissue damage.

### 3.4. LIPUS Protects Gallbladder Motility

To further investigate the mechanism by which LIPUS protects the gallbladder, CCK was utilized to stimulate gallbladder motility, and both in vivo and in vitro experiments were conducted to validate the protective effect of LIPUS. In vivo experimental results indicated that in the NC group, the gallbladder rapidly contracted and emptied after CCK injection (*p* < 0.0001, Figure 4I), whereas the GS group exhibited no distinct contraction. Additionally, results demonstrated that the gallbladder volume in subjects treated with LIPUS rapidly decreased, exhibiting clear contraction, and the gallbladder’s responsiveness to CCK was markedly higher than that of the GS and sham treatment groups (*p* < 0.0001, Figure 4J,L). In contrast, gallbladders in the GS group and the sham treatment group exhibited a minimal response to CCK stimulation (*p* = 0.0835, Figure 4K), characterized by a delayed contraction response and a reduced contraction amplitude. The GBEF in the NC, GS, LIPUS(−), and LIPUS(+) groups were 92.12 ± 7.41%, 30.36 ± 22.79%, 24.84 ± 9.13%, and 83.02 ± 8.50%, respectively. In vitro experiments confirmed the results of the in vivo studies, CCK stimulation of isolated gallbladder muscle strips produced a contraction response higher in the treatment group than in the GS groups (*p* < 0.0001, Figure 4D), although there was no difference between the GS and LIPUS(−) groups (*p* = 0.0835, Figure 4C). The R values in the NC, GS, LIPUS(−), and LIPUS(+) groups were 81.84 ± 14.09%, 23.51 ± 13.38%, 37.74 ± 18.86%, and 89.99 ± 21.25%, respectively. These results indicate that LIPUS effectively protects gallbladder motility by restoring responsiveness to CCK and enhancing contraction strength, which may improve overall gallbladder motility.

### 3.5. LIPUS Restored the Quantity of CCKAR and the Thickness of the Gallbladder Smooth Muscle Layer

In an effort to profoundly study the cause of gallbladder motility restoration by the LIPUS treatment, immunohistochemical staining and Western blot analysis were performed to understand the expression of receptors and proteins related to gallbladder contractibility. The CCKAR receptor antibody was used to label CCKA receptors on smooth muscle cells and Cajal stromal cells, and the number of CCKAR receptors was notably increased in the LIPUS treatment group compared with the LIPUS sham treatment groups (*p* = 0.002, Figure 5A,B). The proportion of CCKAR positive staining for LIPUS(−) vs. LIPUS(+) was 1.07 ± 0.16% vs. 3.69 ± 0.63%. The guinea pig gallbladder smooth muscles were marked by anti-α-SMA antibodies. Compared with the LIPUS(−) group, the thickness of GBSM in the LIPUS(+) group was thickened (*p* = 0.0081, Figure 5C,D).The mean thickness of GBSM in the LIPUS(−) and LIPUS(+) groups were 47.90 ± 7.14 μm and 143.67 ± 33.21 μm. Similarly, in Western blot analysis, we found that the expression levels of CCKAR and α-SMA proteins in the LIPUS treatment group were higher than those in the LIPUS sham treatment group (*p* = 0.0090 and *p* = 0.0041, respectively; Figure 5E–G). The mean grayscale values of the CCKAR protein levels for LIPUS(−) vs. LIPUS(+) were 0.31 ± 0.16 and 1.041 ± 0.53. The mean grayscale values of α-SMA protein levels for LIPUS(−) vs. LIPUS(+) were 0.57 ± 0.07 and 0.850 ± 0.17.

## 4. Discussion

On a high-fat diet, excessive cholesterol accumulates in the walls of the gallbladder [14], which causes chronic inflammation of the gallbladder wall on one hand, and damages the smooth muscle cells and interstitial cells of Cajal on the other hand, leading to a decline in gallbladder contraction function [15], this is an important process in the formation of cholesterol gallstones. This study presents an innovative intervention method to suppress the formation of cholesterol gallstones. By establishing a supersaturated bile acid model in guinea pigs, low-intensity pulsed ultrasound (LIPUS) was used to stimulate their gallbladders, interrupting the early formation stage of cholesterol stones. These experimental findings suggest that LIPUS may effectively interrupt the early phase of cholesterol stone formation by alleviating the damage caused by supersaturated bile to the gallbladder wall and restoring its motility.

As early as 1920, scientists had begun exploring the field of ultrasound therapy. LIPUS has been applied to accelerate the healing of fresh fractures, bone non-union, and soft tissue injury and repair [16]. In recent years, LIPUS has been studied for poorly contracted muscle disorders, such as gastroparesis [17], diabetic erectile dysfunction [18], postpartum urinary retention [3], and so on. LIPUS has been shown to improve muscle contraction, which is consistent with the findings of this study. The efficacy of LIPUS irradiation therapy has been fully confirmed. Although there is still no unified theory regarding its mechanisms, due to differences in methods and parameters used across various studies, numerous studies indicate that LIPUS has therapeutic effects in promoting cell proliferation and differentiation, as well as enhancing tissue anti-inflammatory capabilities [19,20].

Histopathological analysis indicates that, in high-fat diet models, the inflammatory response of the gallbladder wall occurs prior to the formation of cholesterol gallstones. This finding highlights the early pathological changes in the gallbladder wall, suggesting that the inflammatory response may play a notable role in the development of cholesterol stones [21]. The impact of lithogenic bile on gallbladder smooth muscle cells involves alterations in a series of protective mechanisms, leading to edema of the gallbladder wall, increased thickness, and the accumulation of inflammatory cells. Notably, these pathological changes are often accompanied by a decrease in gallbladder contractility and alterations in the gallbladder epithelium’s ability to transport substances, which may affect the normal expulsion of bile and, subsequently, increase the risk of gallstone formation [22]. Additionally, ex vivo analysis of the gallbladder in patients with cholesterol gallstones has shown excess cholesterol in smooth muscle cell membranes, further supporting the important role of cholesterol in the pathophysiological processes of the gallbladder [23]. These results indicate that, after LIPUS exposure, the distribution density of bile salt crystals in the bile clearly decreased, and the inflammatory infiltration and fibrosis of the gallbladder wall were markedly reduced. Furthermore, analysis of the gallbladder’s ultrastructure revealed a decrease in the extent of damage to the gallbladder wall. It may be inferred that LIPUS treatment may improve the contraction ability and material transport capacity of the gallbladder by alleviating inflammation and edema in the gallbladder wall and restoring the function of gallbladder smooth muscle cells.

Previous studies have also demonstrated that LIPUS has positive effects on muscle function as well as cell proliferation and differentiation. In 2016, Ren Yan et al. [3] conducted a clinical trial on LIPUS treatment for postpartum urinary retention, which showed that LIPUS improved the contractility of bladder smooth muscle strips compared to the neostigmine group. Subsequently, Yang B et al. [20] discovered that LIPUS promoted the recovery of bladder leak points in a rat model of stress urinary incontinence; their study found an increase in the myogenic differentiation level of the urethra and the phosphorylation level of p38, supporting a correlation between the two. In 2023, research indicated that LIPUS regulated macrophage polarization towards an anti-inflammatory phenotype by activating the WNT signaling pathway, thereby promoting skeletal muscle regeneration [24]. In our study, we observed an increase in α-SMA protein expression along with thickening of the gallbladder smooth muscle layer following LIPUS treatment. This may also assist in enhancing gallbladder contraction, to some extent. However, the specific mechanisms remain unknown and warrant further investigation.

CCKAR is the primary physiological mediator of gallbladder smooth muscle contraction. CCK can induce gallbladder smooth muscle contraction by acting directly on CCKAR in the gallbladder. The reduced levels of gallbladder CCKAR lead to decreased gallbladder motility, weakened gallbladder contractions, and promote the formation of gallstones [25]. One-third of mice with CCKAR deficiency spontaneously developed gallstone disease at 12 months and 24 months of age. Studies have shown that feeding mice a lithogenic diet can decrease the expression of CCKAR in the gallbladder. Due to the absence of the CCKAR receptor in mice, this results in cholesterol gallstone formation due to gallbladder stasis. A previous study reported that the total expression of CCKAR protein in the gallbladders of mice fed a high-fat diet for 1 month was distinctly reduced compared to control mice. Another study observed that Caveolin-3 (CAV3) is involved in regulating the diminishment of gallbladder muscle motility, suggesting that CAV3 and CCKAR are implicated in cholesterol gallstone formation, as lithogenic diets rich in cholesterol reduce CCK-induced contractions by sequestering CCKAR in the caveolae of gallbladder smooth muscle cells. Consistently, the results of this study indicated that, after receiving the LIPUS treatment, the impaired gallbladder showed outstanding increased responsiveness to CCK. Connexion43 (CX43) is an important gap connexin, which regulates signal communication between cells and the transmembrane transport of ions. Previous research has shown that CX43 protein expression in guinea pigs decreased after 8 weeks of consuming a high-fat diet [5]; thus, gallbladder contractility was injured. Our TEM results showed that LIPUS may regulate CX43 expression to some extent, but this needs further experimental research. Immunohistochemical and Western blot analyses confirmed that LIPUS treatment restored the quantity of CCKAR on the surface of smooth muscle cells and interstitial cells of Cajal, while also thickening the smooth muscle layer. The effect may help to recover the gallbladder’s poor contraction ability caused by a high-fat diet to some extent.

In this study, the changes in bile crystal density, gallbladder wall inflammation, CCKAR density, and muscle thickness during the application of LIPUS in cholesterol-induced cholelithiasis were investigated, we preliminarily demonstrated the therapeutic potential of LIPUS for cholesterol-induced cholelithiasis. One limitation of this study lies in that our study failed to use homogeneous animal models with male and female animals for preclinical study. This may lead to findings that are not applicable across different sexes. As well, we did not do in-depth research to clarify the specific molecular mechanisms of the changes in the CCKAR and gallbladder muscle layer. In addition, our research is limited to the effect of LIPUS on gallbladder diseases, we will expand the application and research of LIPUS in other biliary and gastrointestinal motility disorders in the future. For instance, improving muscle function would be beneficial for dysfunction in the sphincter of Oddi, also known as biliary dyskinesia [26], and achalasia of cardia, which is an esophageal motility disorder characterized by ineffective peristalsis of the smooth muscle of the esophageal body and relaxation of the lower esophageal sphincter [27]. However, it must be mentioned that the function of LIPUS requires a certain timeliness. Therefore, it cannot be used for severe acute conditions, such as acute cholelithiasis.

## 5. Conclusions

In this study, we successfully constructed a cholesterol gallstone disease model in a guinea pig and used it as an experimental subject to study the alterations of gallbladder cells, tissues, and organs under LIPUS irradiation. The results showed that LIPUS irradiation was able to improve the gallbladder contractility and regulate the inflammatory environment of the gallbladder wall in guinea pigs with cholesterol gallstone disease. The mechanism of the enhancement of the contractility in the gallbladder could be related to the fact that LIPUS restored both the quantity of CCKAR and the thickness of the gallbladder smooth muscle layer.

## Figures and Tables

**Figure 1 bioengineering-12-00034-f001:**
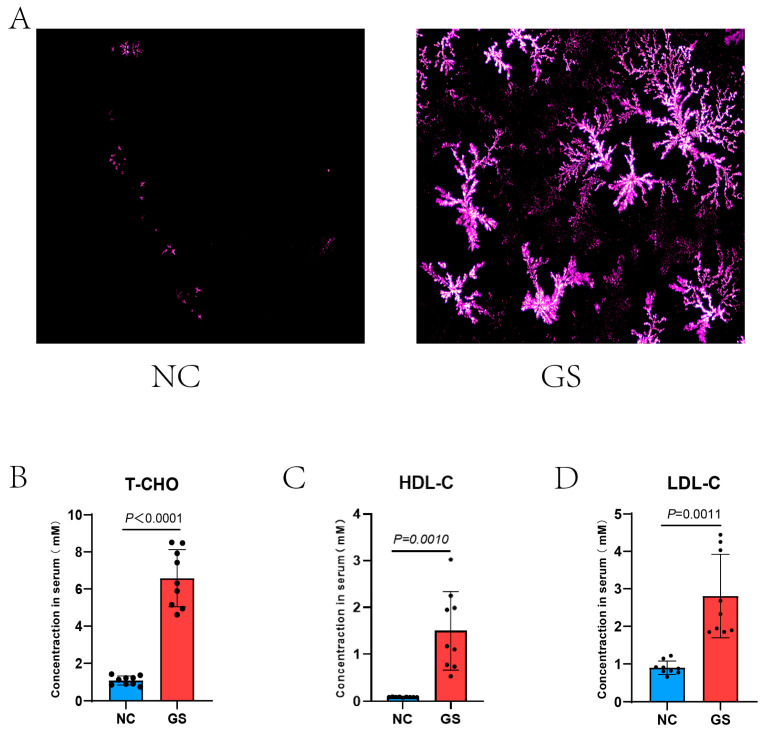
(**A**) Bile crystals are polygonal in shape and their density was visibly elevated after 8 weeks on the lithogenic diet compared with the NC groups; little bile crystals are visibly scattered. (**B**–**D**) Serum T-CHO, HDL-C, and LDL-C values of NC vs. GS: T-CHO: 1.09 ± 0.24 mM vs. 6.58 ± 1.54 mM, HDL-C: 0.09 ± 0.01 mM vs. 1.50 ± 0.83 mM, and LDL-C: 0.90 ± 0.18 mM vs. 2.81 ± 1.11 mM with significant differences of *p* < 0.0001, *p* = 0.0010, and *p* = 0.0011, respectively. A two-tailed Student’s *t*-test was used for statistical analysis. GS: gallstone group; NC: normal control.

**Figure 2 bioengineering-12-00034-f002:**
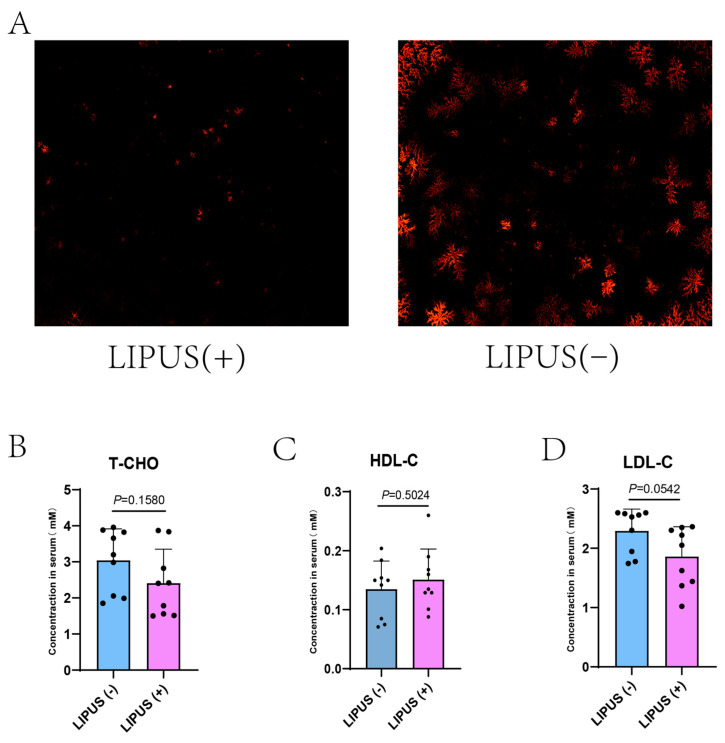
(**A**) The density of bile salt crystals (magnification at 50×) in the treatment group was statistically significantly lower than that in the sham group. (**B**–**D**) Serum T-CHO, HDL-C, and LDL-C values for LIPUS(+) vs. LIPUS(−): T-CHO: 2.41 ± 0.94 mM vs. 3.04 ± 0.87 mM, HDL-C: 0.15 ± 0.05 mM vs. 0.13 ± 0.05 mM, and LDL-C: 2.293 ± 0.50 mM vs. 1.86 ± 0.37 mM with *p* values of *p* = 0.1580, *p* = 0.0542, *p* = 0.5024, respectively. A two-tailed Student’s *t*-test was used for statistical analysis.

**Figure 3 bioengineering-12-00034-f003:**
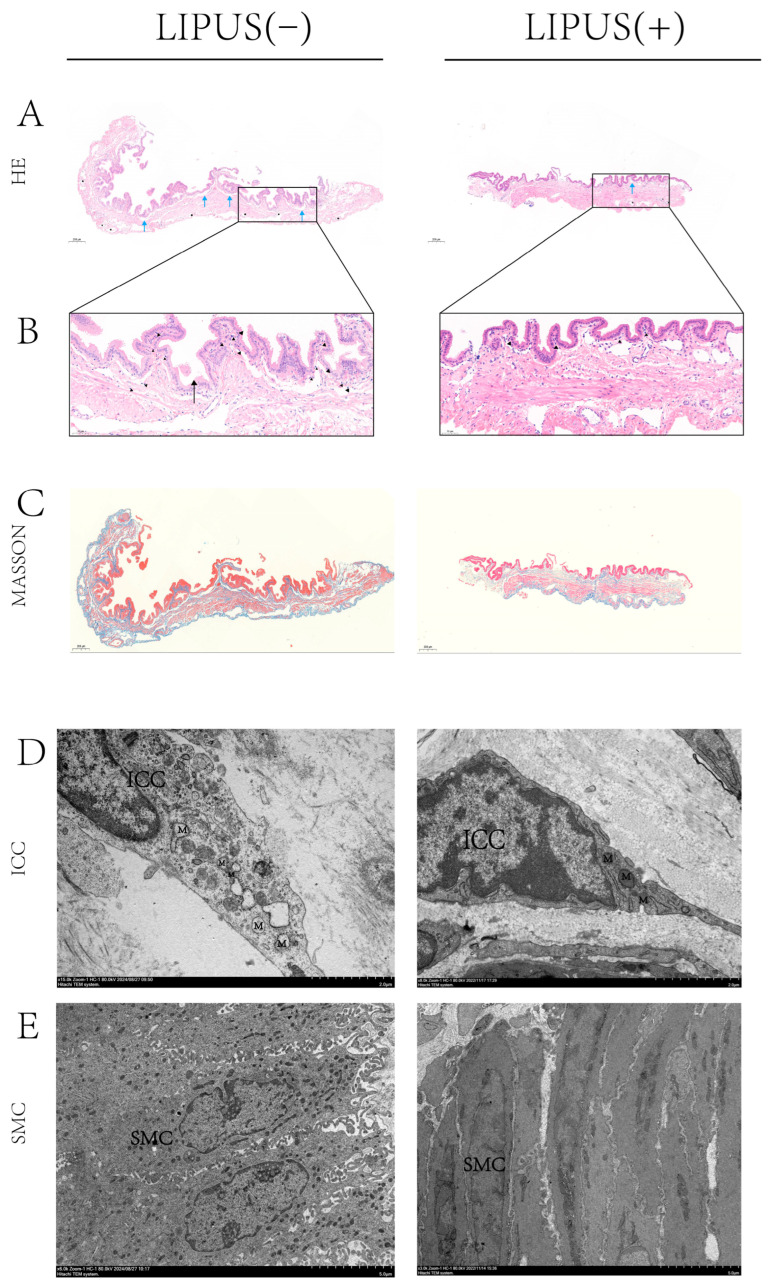
(**A**,**B**) Histopathological analysis of gallbladder tissue inflammation at 200× magnification; the inflammatory infiltration of the gallbladder wall was reduced in the treatment group, the infiltration of neutrophil cells (▲) was reduced, tissue edema (→) was improved, and new capillaries (★) were reduced. In addition, the R-A sinus (→) was present in the sham group. (**C**) Gallbladder Masson staining at 200× magnification shows there is no difference in gallbladder wall fibrosis between the treatment group and the sham group. (**D**,**E**) TEM observation of the ICC and SMC structures in guinea pig gallbladders. In the sham group, mitochondria within the ICCs of the gallbladder wall exhibited irregular swelling, the endoplasmic reticulum was markedly swollen, and the nuclear membrane displayed signs of contraction (**D**, left, 15k×). The morphology of SMCs exhibited deformation, characterized by sparse synaptic structures and the nearly complete disappearance of intracellular reticular structures (**E**, left, 6k×). M: mitochondrion, ICC: interstitial cell of Cajal, SMC: smooth muscle cell, R-A: Rokitansky–Aschoff. “Appendix A” for pathological figure details.

**Figure 4 bioengineering-12-00034-f004:**
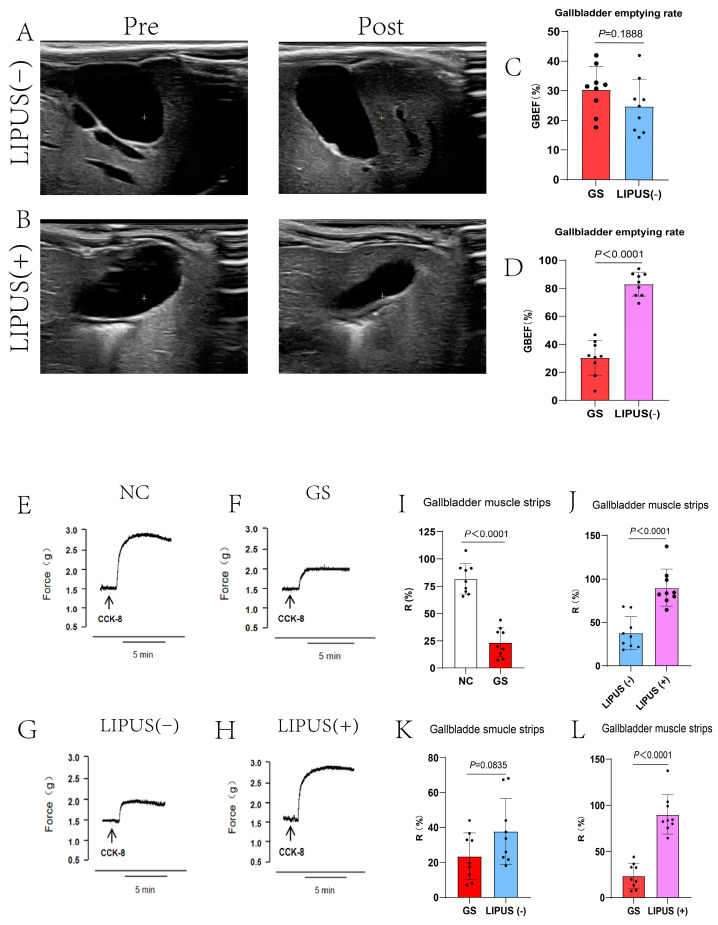
(**A**,**B**) Ultrasound evaluation of guinea pig gallbladder contractibility. Pre-injection of CCK-8 (**A**,**B** left) and post-injection of CCK-8 (**A**,**B** right). Gallbladders in the GS group and the sham treatment group exhibited a minimal response to CCK-8 stimulation (*p* = 0.1888 vs. GS (**C**)), whereas the gallbladder rapidly contracted and emptied after CCK-8 injection in the LIPUS(+) group (*p* < 0.0001 vs. GS (**D**)). GBEF in the NC, GS, LIPUS(−), and LIPUS(+) groups were, respectively, 92.12 ± 7.41%, 30.36 ± 22.79%, 24.84 ± 9.13%, and 3.02 ± 8.50%. (**E**–**H**) Reactivity curve of guinea pig gallbladder smooth muscle strips to CCK-8. The R values in the NC, GS, LIPUS(−), and LIPUS(+) groups were 81.84 ± 14.09%, 23.51 ± 13.38%, 37.74 ± 18.86%, and 89.99 ± 21.25%, respectively, with corresponding *p* values of *p* < 0.0001 vs. NC (**I**), *p* < 0.0001 vs. LIPUS(−) (**J**), *p* = 0.0835 vs. GS (**K**), and *p* < 0.0001 vs. GS (**L**). A two-tailed Student’s *t*-test was used for statistical analysis.

**Figure 5 bioengineering-12-00034-f005:**
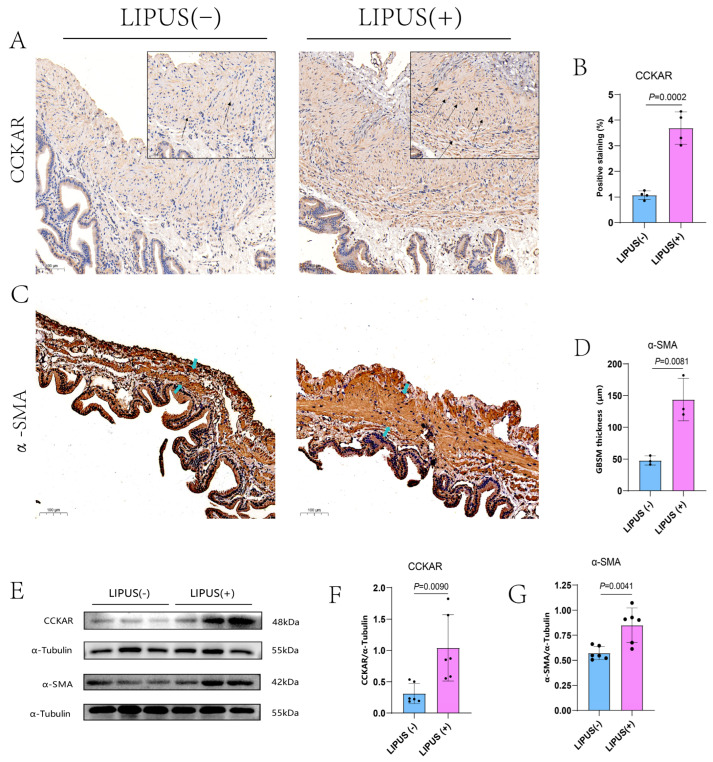
(**A**) The black arrow shows the CCKAR immunohistochemically positive area of the gallbladder (scale of bar: 100 μm), which was intensively expressed in the LIPUS treatment group (**A** right). (**B**) The proportion of CCKAR-positive staining for LIPUS(−) and LIPUS(+) was 1.07 ± 0.16% and 3.69 ± 0.63%, *p* = 0.002 vs. LIPUS(−). (**C**) The blue arrow indicates that the continuous deep staining area (scale of bar: 100 μm), which is the smooth muscle layer with positive staining, is clearly thickened in the LIPUS treatment group (**C** right). (**D**) The mean thickness of the GBSM in the LIPUS(−) and LIPUS(+) groups was 47.90 ± 7.14 μm and 143.67 ± 33.21 μm, *p* = 0.0081 vs. LIPUS(−). (**E**) The protein expression level of CCKAR and α-SMA increased after LIPUS treatment in the CGS model of guinea pigs. (**F**) The mean grayscale values of CCKAR protein levels in the LIPUS(−) and LIPUS(+) groups were 0.31 ± 0.16 and 1.041 ± 0.53, *p* < 0.0090 vs. LIPUS(−). (**G**) The mean grayscale values of α-SMA protein levels in the LIPUS(−) and LIPUS(+) groups were 0.57 ± 0.07 and 0.850 ± 0.17, *p* = 0.0041 vs. LIPUS(−). A two-tailed Student’s *t*-test was used for statistical analysis.

## Data Availability

Data from the current study are available from the corresponding author upon reasonable request.

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
