# Peer review of "The Therapeutic Potential of Low-Intensity Pulsed Ultrasound in Enhancing Gallbladder Function and Reducing Inflammation in Cholesterol Gallstone Disease"

_bioengineering, 2025, doi:10.3390/bioengineering12010034_

Round 1

Reviewer 1 Report

Comments and Suggestions for Authors

The authors investigated the effects of low intensity pulsed ultrasound in a gall stone model in guinea pigs. They found that its use resulted in higher gall bladder contractility and bile clearance and was associated to less inflammation.

1. The authors used only male guinea pigs, although we know that sex-specific experiments are less likely to be reproducible and transferrable.

2. Standard error of the mean is not the adequate measure of variability. The use of an experimental group with an animal being the unit of interest requires the use of the standard deviation.

3. ANOVA and t-test are less susceptible to a violation of the normality assumption than on the violation of the homogeneity of variances. How has the latter been assessed?

4. Besides remark #3, ANOVA requires the residuals to be normally distributed, not the data.

5. Presenting results with just p-values is inacceptable, as they are nothing without data. Moreover, please present exact p-values and not whether they are below or above an artificial threshold.

6. In the figures, it is unclear which test has been used and how the data are presented. In addition, please include individual values if you prefer a bar graph.

7. If significantly in the text means statistically significant, this needs to be termed as such because it is a terminus technicus.

8. The use of abbreviations is inconsistent. For example: In line 161, the abbreviation is the subtitle, which gets first explained in line 226 while it had also been used in line 170 as an abbreviation or CAV3 in line 365 is not explained at all.

9. Panels C to L are barely legible in Figure 4, so it might be preferrable to have them as another figure. To a lesser extent, this is also the case for figure 5C/D/F/G.

10. The authors have not provided the reader with manufacturer names for materials except for the ultrasound machine and the staining antibodies.

11. In the current size and resolution, the histology images are difficult to view, too.

12. The discussion does not address the rather artificial aspect of gall bladder contractility beyond the model. Has this any relevance in gallstone formation beside the current model? Is there any evidence for such a pathophysiology that would be altered by such an approach or is this purely mechanistic without relevance for the human disease mentioned in the introduction?

Reviewer 2 Report

Comments and Suggestions for Authors

This research report investigates and reports the effects of low-intensity pulsed ultrasound (LIPUS) on improving gallbladder contractile function and reducing chronic inflammation in cholelithiasis (CGS). Using animal models, they report that LIPUS treatment restored gallbladder smooth muscle thickness and CCK receptor (CCKAR) expression and improved gallbladder motility. In addition, accelerated bile crystal discharge, reduced inflammation, and formation of new capillaries have been observed, and inhibition of CGS progression has been observed. Such demonstration of the usefulness of LIPUS for patients suffering from cholelithiasis is useful, and its clinical application is expected in the future.

On the other hand, there are some points to be pointed out, please add the following points to the discussion.

1.                 This is a study in a model animal. Discuss possible future applications for human clinical trials.

2.                 The LIPUS treatment setting is different for each patient in humans. Please consider the possibility of applying the results of this study to human clinical practice.

3.           Discuss the long-term safety and efficacy of the product.

4.           Applicability to patients with severe CGS and comorbidities is unknown.

5.           We feel that there is a lack of comparative validation with existing treatments. Many approaches have been approached to treat this disease. Please explain where the usefulness of this time lies, for example, it will be applicable to children, the elderly, and patients with chronic diseases.

Reviewer 3 Report

Comments and Suggestions for Authors

The manuscript titled “The Therapeutic Potential of Low-Intensity Pulsed Ultrasound in Enhancing Gallbladder Function and Reducing Inflammation in Cholesterol Gallstone Disease” investigates an innovative and non-invasive approach using low-intensity pulsed ultrasound (LIPUS) to address gallstone-related dysfunctions. The study establishes a solid experimental model and presents compelling results showing that LIPUS can improve gallbladder contractility, reduce inflammation, and mitigate damage associated with cholesterol gallstones. The findings are significant and contribute to the field of non-pharmacological treatments for gallstone disease. However, the manuscript requires several revisions to improve its clarity, rigor, and comprehensiveness.

Suggestions for Major Revisions:

1. Clarification of Mechanisms:

While the study mentions the beneficial effects of LIPUS on smooth muscle cells and inflammation, the molecular mechanisms remain insufficiently detailed. I recommend including a more comprehensive discussion on the non-thermal effects of LIPUS and its impact on cellular and molecular pathways, such as the modulation of cholecystokinin A receptor (CCKAR) and α-SMA expression.

2. Experimental Design and Statistical Analysis:

The rationale for the statistical methods, such as the use of Mann-Whitney or Kruskal-Wallis tests, is not adequately justified. Please provide a more detailed explanation of why these methods were chosen and include effect sizes along with p-values to strengthen the statistical robustness of the results.

3. Control Groups:

The current experimental design lacks additional relevant control groups, such as a group receiving pharmacological treatment for gallstones or dietary interventions. Adding these groups could provide a better context for evaluating the efficacy of LIPUS as a standalone treatment.

4. Data Presentation:

Some figures (e.g., Figures 2 and 3) require more detailed legends and clearer labeling of group comparisons. Additionally, quantitative data (e.g., protein expression levels in Figure 5) should include standard error bars and statistical significance annotations to ensure transparency.

5. Literature Review and References:

The introduction and discussion sections rely heavily on older references, which may not reflect the latest advancements in LIPUS research and gallbladder physiology. Updating the citations with recent studies will enhance the manuscript’s credibility and relevance.

6. Broader Implications in the Discussion:

The discussion is narrowly focused on gallstone disease. Expanding on the potential applications of LIPUS in other biliary conditions or gastrointestinal disorders could broaden the study’s clinical relevance and impact.

These revisions will significantly enhance the clarity and scientific rigor of the manuscript, ensuring that the findings can be better appreciated by readers.

Round 2

Reviewer 1 Report

Comments and Suggestions for Authors

The authors provide a revised version of their manuscript, which has been substantially improved. However, somme issues persist. The enumeration of my comments pertains to the previous review. If I do not address the number, the issue has been completely resolved by this revision.

1. I am afraid that I cannot accept that explanation. Aren't the authors aware of the work by Kline & Karpinski in Physiological Reports 2016;4:e12843, which had already shown that sex is not a relevant factor in gall bladder motility? Please explain.

In addition, this is a relevant draw-back (see for example this piece by Karp & Reavy in the British Journal of Pharmacology 2019;176:4107-4118), which requires discussion of the lack of generalisability of the results and an increased risk of non-reproducibility for sex-specific preclinical research.

3.-4. Thank you for providing the results of the tests, but please include the description of the conducted tests in the methods and take into account the issue with the Gaussian distribution of the residuals in the text in section 2.11, too. Please note that these comparisons have to be conducted for each variable, but I assume that this has been done and you did only provide us with an example, am I correct?

5. While the data is now provided alongside the exact p-values, I may suggest to place the p-value behind its comparison to increase the understandability for the reader throughout the text. This would especially be relevant for lines 271, 274, and 276.

6. Thank you very much for offering to provide the raw data. I appreciate this openness and would of course encourage the authors to make the raw data publicly availabe, but for the sake of this review, providing individual data points is sufficient.

What is still missing in the figure legends is the information which statistical test has been used for the comparison. Please add this to the legends.

8. In the revised version, the authors introduced again some abbreviations that were not mentioned before and never afterwards like RIPA in line 151, BCA in line 152 or PVDF in line 155. I may suggest to not introduce an abbreviation if you do not intend it to use a second time in the text, as it is for example the case with BSA in line 142, HRP in line 143 or DAB in line 145.

10. You describe that the LIPUS machine was sponsored. Can you please comment on any influence the manufacturer had on your study at any point? If there was none, please state accordingly in the declaration and include this as funding information for transparency.

Otherwise, well done.

Reviewer 2 Report

Comments and Suggestions for Authors

The authors have made appropriate corrections and additions to my previous comments. I believe that this paper will provide much more useful information to readers than before. I have no further comments to make.

Author Response

Thank you very much for your review and correction. 

Reviewer 3 Report

Comments and Suggestions for Authors

I have no further suggestions for authors, I feel that the paperhas been well revised.

Author Response

(The authors gave the same response as above.)

Round 3

Reviewer 1 Report

Comments and Suggestions for Authors

The authors provide a revised version of their manuscript. Except for the issue of sex-specific research animals, my comments have sufficiently been addressed. In order to make it easier for the authors, I may suggest a specific sentence how to address that issue.

"Male animals were selected to minimize the effect of sex-specific estrogen on experimentment cholesterol gallstone formation." (lines 70-72)

Please change to

Male animals were selected with the intend to minimize the effect of sex-specific estrogen on experimentment cholesterol gallstone formation, as we were unaware of a previous study that demonstrated no influence of sex-hormones on gallbladder motility [citation to Kline & Karpinski].

I feel that one cannot know everything, but it will certainly beneficial for the field if that work by Klin & Karpinski gets more known and we will not see sex-specific research anymore. For full transparency, I am not an author on that paper nor is it from any of my collaborators or my institution.

The new statement in lines 352-354 is not backed by data from this study or a citation. Please remove as it is massively overstated.
